# Extracellular Vesicle Metabolomics Holds Promise for Adult Axon Regeneration

**DOI:** 10.3390/metabo15070454

**Published:** 2025-07-04

**Authors:** Maria D. Cabrera Gonzalez, Jackson Watson, Laura Leal, Isabella Moceri, Camille Plummer, Biraj Mahato, Abdelrahman Y. Fouda, Sanjoy K. Bhattacharya

**Affiliations:** 1Miami Integrative Metabolomics Research Center, Bascom Palmer Eye Institute, University of Miami, Miami, FL 33136, USA; mdc209@med.miami.edu (M.D.C.G.); jxw1987@miami.edu (J.W.); igm243@med.miami.edu (I.M.); cxp3042@med.miami.edu (C.P.); 2Department of Biochemistry and Molecular Biology, University of Miami, Miami, FL 33136, USA; 3Department of Medicine, Universidade Positivo, Curitiba 81280-330, Paraná, Brazil; lauraleal1405@gmail.com; 4Children’s Hospital Los Angeles, University of Southern California, Los Angeles, CA 90027, USA; bmahato@chla.usc.edu; 5Department of Pharmacology and Toxicology, University of Arkansas for Medical Sciences, Little Rock, AR 72205, USA; afouda@uams.edu

**Keywords:** extracellular vesicles, exosomes, metabolomics, metabolites, lipids, mass spectrometry, multi-omics, adult axonal regeneration, retina ganglion cells, microglia, optic nerve

## Abstract

Extracellular vesicles (EVs) are bilayer lipid membrane particles that are released by every cell type. These secretions are further classified as exosomes, ectosomes, and microvesicles. They contain biomolecules (RNAs, proteins, metabolites, and lipids) with the ability to modulate various biological processes and have been shown to play a role in intercellular communication and cellular rejuvenation. Various studies suggest exosomes and/or microvesicles as a potential platform for drug delivery. EVs may deliver lipids and nucleotides directly to an injury site in an axon, promoting growth cone stabilization and membrane expansion as well as repair, thus positively modulating adult axon regeneration. In this review, we will provide a perspective on the metabolite composition of EVs in adult axonal regeneration relevant to the central nervous system (CNS), specifically that pertaining to the optic nerve. We will present an overview of the methods for isolation, enrichment, omics data analysis and quantification of extracellular vesicles with the goal of providing direction for future studies relevant to axon regeneration. We will also include current resources for multi-omics data integration relevant to extracellular vesicles from diverse cell types.

## 1. Introduction

Axonal injuries represent a significant clinical and biological challenge, often leading to permanent functional deficits due to the limited regenerative capacity of central nervous system (CNS) axons and the partial potential recovery in the peripheral nervous system (PNS) [1,2]. The interaction between intrinsic neuronal mechanisms (within the neurons) and extrinsic environmental factors (in interstitial spaces) supports this variability in regenerative outcomes [3,4]. Among the emerging mediators of axonal repair, EVs– specifically exosomes– have garnered attention for their capacity to act as intercellular messengers, delivering bioactive molecules (metabolites, RNAs, proteins, and lipids) that influence neuronal survival and inflammation [5,6].

Metabolomics, the comprehensive study of small-molecule metabolites, provides a powerful lens to explore the molecular intricacies of EV-mediated axonal regeneration [7]. By profiling the metabolic signatures of exosome cargo, researchers can uncover the biochemical pathways that drive repair processes. However, several critical aspects still require clarification. Identifying the key metabolic profiles of EVs that optimize neuroregeneration is essential for advancing the field, providing insight into how injury-specific conditions influence the composition of the exosome metabolome [2]. Furthermore, translating this knowledge into effective therapeutic interventions holds significant potential for improving regenerative outcomes both in the CNS and PNS.

This review consolidates the current understanding of exosome-mediated axonal regeneration and examines the promise of metabolomic approaches in advancing this field. By compiling established isolation and characterization methods, we aim to highlight opportunities for leveraging exosome metabolomics to decode the molecular basis of axonal repair and to develop innovative regenerative therapies.

## 2. Axon Regeneration

Adult axonal regeneration is a highly complex mechanism due to the lack of plasticity among neurons and the environmental cues that might limit regeneration depending on the site of injury [8,9,10,11,12]. Neurons are asymmetrical units of the nervous system that consist of a soma (made of the cell’s organelles and genetic material), dendrites (receive electrical and chemical signals from other cells), and axon (aids transmission of electrical signals to other cells) [13]. The asymmetry is manifested by the differences in lengths between dendrites and axons. Axon myelination facilitates the rapid transmission of electrical signals to cells in every part of the body [13]. Injuries to the nervous system led to the possible destruction of an axon both in the PNS and CNS. Nerve injuries trigger a stereotypical cascade termed Wallerian degeneration [14,15]. During this process, clearance of myelin and axons happens distal to the injury by macrophages [16]. This process differs greatly in the PNS and CNS. In the PNS, axons have demonstrated a higher capacity to regenerate compared to those in the CNS [11]. This phenomenon is due to the presence of oligodendrocytes instead of Schwann cells in the CNS, environmental differences, and immune responses. Insults to the PNS have been shown to elicit a robust immune response from both neuronal and non-neuronal cells, contributing to heightened regeneration by the elevated presence of macrophages in the injury site [16]. Rapid remyelination from Schwann cells allows for axonal regeneration by providing the necessary growth factors and cellular environment to promote regrowth.

Adult axonal regeneration within the CNS remains a fundamental challenge in neurobiology [17]. Unlike during the development stage, when axons actively extend and establish new synaptic connections despite the changes in their environment, the adult CNS exhibits minimal spontaneous regrowth following injury [8,9,10,12,18,19]. Among CNS structures, the optic nerve serves as a widely used model for studying axonal regeneration due to its accessibility, well-characterized anatomy, and relevance to human diseases such as glaucoma and optic neuropathies [20].

The failure of axonal regeneration in the CNS stems from both intrinsic neuronal limitations and a highly inhibitory extracellular environment [21]. Retinal ganglion cells (RGCs), which give rise to the optic nerve, undergo a developmental downregulation of growth-associated genes, leading to a diminished intrinsic capacity for axonal elongation [22]. Simultaneously, the post-injury environment becomes highly inhibitory, with myelin-associated molecules, reactive astrocytes, and glial scarring all contributing to a non-permissive landscape [23,24,25]. To address these barriers, experimental strategies have targeted the reactivation of intrinsic growth pathways—such as the mTOR and JAK/STAT cascades—while also seeking to modulate the extracellular milieu [4,26,27]. In this context, EVs have emerged as promising candidates for CNS repair [3,28].

These nanoscale, membrane-bound structures carry a diverse cargo of proteins, lipids, and non-coding RNAs that can modulate recipient cell behavior and influence regenerative signaling cascades. Extracellular vesicles are membrane-bound structures released by cells across all domains of life, ranging from 30 nm to over 1 µm in size [6,29]. These particles are further classified based on their site of origin and size. Based on the classification provided by the International Society of Extracellular Vesicles, an exosome has an endosomal origin, while microvesicles (also known as ectosomes) are plasma membrane-derived [6]. EVs are also classified as “small EVs” or “medium EVs”, but in this paper, we focus on the role of exosomes specifically [6]. The size of EVs encompasses different ranges—exosomes are usually less than 200 nm, microvesicles are 100–1000 nm and apoptotic and inclusion bodies are usually 50–5000 nm (Figure 1). The historically well-characterized subtypes have been presented in Figure 1. This field is rapidly advancing, and several new categories of EVs, such as oncosomes, exomeres, and supermeres, are emerging.

In models of optic nerve injury, EVs derived from mesenchymal stem cells and neural progenitor cells have been shown to reduce neuroinflammation, promote axonal growth, and enhance neuronal survival [30]. Their small size, biological stability, and capacity for targeted delivery make them particularly attractive for therapeutic applications. Furthermore, EVs have shown the potential to induce and maintain repair phenotypes in glial cells and promote neurite outgrowth through the modulation of molecular pathways critically implicated in axon regeneration [31,32]. Together, these insights signal a paradigm shift—from passive observation of regenerative failure to active modulation of the cellular and molecular landscape underlying CNS repair [32]. As omics technologies and vesicle engineering continue to advance, EV-based strategies may offer innovative and clinically translatable approaches for restoring vision and function following optic nerve injury [32].

Recent advances have illuminated the role of exosomes in axonal repair. These vesicles are enriched with critical molecular cargo, such as miRNA-21, which regulates pathways such as PTEN-PI3K signaling, driving axonal growth and neuronal recovery [4,26,30,33]. Despite these breakthroughs, the mechanisms governing EV cargo selection, release, and targeted delivery remain incompletely understood. Further research is required to establish a holistic understanding of EV metabolomics, lipidomics and proteomics. Contributing to implementing possible therapeutics for axon regeneration using EVs.

## 3. Isolation

The purification of extracellular vesicle extraction from the retina consists of extraction of the retina isolation of retinal ganglion cells and microglial cells, followed by the purification of extracellular vesicles from those cells [34]. RGCS are of interest because they are the primary neurons whose axons bundle to form the optic nerve. The retina is isolated from the enucleated eye through a surgical technique that separates the cornea from the rest of the eyecup. Forceps then are used to slide the retina, residual pigment epithelium, and lens out of the eyecup. Finally, the retina is extracted from the collection and placed by itself using forceps [35]. Different techniques are used to achieve this isolation, such as using a needle to puncture the cornea of the eye and creating a flat retina flap [36,37].

To enrich the extracellular vesicles of note, isolation of the RGCs is necessary. One study provided three distinct ways to isolate the RGCs where each method had no effect on the efficiency of cell recovery [36]. The step for each method places up to 12 retinas on a sterile 70 µm nylon moistened strainer. The first method uses the back end of a 10 mL syringe to gently macerate the retinae using a circular motion [36]. A second method uses a pestle to macerate the cells, and the third uses enzymatic digestion with a combination of papain, L-cysteine, and DNase I, followed by inactivation of the solution [36]. After each method is performed, the nylon strainer is placed over a polypropylene collection tube. The cells are passed through the strainer using a P1000 pipette. The strainer is then rinsed to release any remaining cells and is once again transferred to the collection tube. More solvent is added to achieve a final volume of 1 mL per retina and is then centrifuged to form a pellet [36]. After the isolation of cells, multiple distinct methods may be employed that enable the isolation of the extracellular vesicles and exosomes from the RGCs, microglial cells and other media (Figure 2).

No single method enables optimal isolation of EVs and exosomes from cells. However, the combined usage of multiple methods has been shown to prove effective, allowing for the maximal purification of exosomes from samples [6]. Commercial kits such as “Total isolation from cell culture” by ThermoFisher and “ExoQuick^®^ Exosome Isolation and RNA purification kit” by system bioscience are popular choices by researchers as a first method of purification. A variety of kits are commercially available for exosome isolation (Table 1). Protocols will usually follow the use of these kits with ultracentrifugation, precipitation and other methods discussed below (Figure 2).

Neural progenitor cell isolation from mice requires excision of the brain. After excision, the subventricular zone is dissected. The chunks are then centrifuged and resuspended multiple times in a NPC medium with the addition of growth factors after day 3 [38]. These cells were then enriched for EVs. Mesenchymal stem cells are obtained from adipose tissue. This tissue is digested with a collagenase and cultured. The following cells are incubated in MSC culture medium until further EV isolation [38].

**Table 1 metabolites-15-00454-t001:** Isolation kits for extracellular vesicles.

Isolation Kits			
Company	Type of Isolation	Catalog #	Reference
ThermoFisher (Waltham, MA, USA)	Total Isolation from cell culture	4478359	[39]
	Total Isolation from plasma	4484450	[40,41]
	Total Isolation from serum	4478360	[39]
	Total Isolation from urine	4484452	[39]
	Total Exosome RNA and Protein Isolation	4478545	[39]
QIAGEN (Hilden, Germany)	exoEasy Maxi Kit	76064	[42]
	miRCURY Exosome Serum/Plasma Kit	76603	[42]
System bioscience (Palo Alto, CA, USA)	ExoQuick-TC^®^ ULTRA for Tissue Culture Media	EQULTRA-20TC-1	[43,44]
	The Original ExoQuick	EXOQ5A-1	[44]
	ExoQuick^®^ Exosome Isolation and RNA Purification Kit (for Tissue Culture Media)	EQ806TC-1	[43]
	ExoQuick^®^ Exosome Isolation and RNA Purification Kit (for Serum and Plasma)	EQ806A-1	[41]
IZON (Christchurch, New Zealand)	qEVoriginal Columns	ICO-35	[42,44]
Fujifilm (Tokyo, Japan)	MagCapture ™ Exosome Isolation Kit PS Ver.2	294-84101	[45,46]

A comparative proteomics study performed by ThermoFisher Scientific compared six serum exosome procedures. It demonstrated that the Total Exosome Isolation kit identified the least amount of protein groups in comparison to the other kits. The highest exosome purity was obtained by using a Mag Capture kit, but it did provide the lowest protein yield.

Precipitation is a commonly used method for isolating exosomes from media, allowing a solution to wrap the exosomes. The optimized solution is polyethylene glycol (PEG), as it has been shown to wrap exosomes, creating a film that collects into a supernatant under centrifugation (Figure 2C) [37]. The precipitation approach is cost-efficient, simple, and quick, requiring only a centrifuge and a syringe to separate supernatant from media. This method is great for attaining a supernatant composed of exosomes; however, the non-vesicular extracellular proteins, immunoglobins, and other molecules can aggregate in the supernatant as well [47]. In a worldwide survey in 2015, 14% of researchers used precipitation techniques in their isolation of EVs [48]. This is an indication that while precipitation can be simple and effective, it does not ascertain pure isolations of extracellular vesicles.

Differential ultracentrifugation separates particles based on their buoyant densities and sedimentation properties. Centrifugation causes particles that are heavier than the solvent to collect into sediment (Figure 2D) [49]. A major limitation of this method is its inability to accurately separate small disparities in particle sedimentation rate (e.g., small EVs, large EVs and extracellular proteins). One way to address the issues that can arise with co-precipitation of other media like apoptotic bodies is a sucrose gradient, which separates particles based on their buoyant densities [50]. This density separation can separate nanoparticles like exosomes from retinal ganglion cells or other cell media; however, products like lipoproteins and HDLs have a similar density and size to exosomes, causing them to aggregate together [51,52]. The differential ultracentrifugation is best used as a “cut-off size”—based protocol [49]. While differential ultracentrifugation might not be able to separate distinct EVs by their size, it is suitable and is considered a “gold standard” for isolating extracellular vesicles from diverse cell media [53].

Immunoaffinity (IA) capture is an isolation method using antibodies or magnetic properties to isolate exosomes from cell media (Figure 2F) [54]. The targeted EVs present shared as well as distinctly different extracellular proteins. While it is necessary to have classifications of proteins that are different between targeted EVs, these markers, known as “exosome markers,” can be used to isolate sets of EVs by affinity to their antibodies [55]. These markers are limited by their influence on distinct EV populations. They differ greatly by cell culture, and therefore, this method can be inconsistent without preliminary marker testing. The antibodies may be assembled into a membrane or onto magnetic beads and centrifuged. When compared to centrifugation and density-based separation, IA capture was able to enrich exosomes by at least twofold more [56]. However, IA capture can be difficult to knock the EVs from the beads or membranes, with extra treatments necessary to separate the antibodies from the EVs [6]. This technique is expensive and requires careful preparation of antibodies and magnetic beads. On the other hand, this is a highly specific method, yielding pure and specific exosomes, and it can be easily scaled to larger sample volumes [57]. It is important to note that IA is always used in conjunction with another isolation technique, such as some form of prior capture of particles [58].

Size exclusion chromatography (SEC) is becoming a very popular method for EV fractionation due to its simplicity, feasibility, and retained function of EVs [59]. SEC separates nanoparticles based on size using a vertical column with a matrix that has a defined, even pore size (Figure 2G) [60,61]. It is usually gravity-driven, but pressure delivered by the pumps has also been used for efficient and finer separations. Some advantages of SEC are the negation of risk for protein complex formation and vesicle aggregation by centrifugation methods [52]. The EVs can also be isolated from proteins and HDL, but it is not as pure as other methods [52]. The small EVs/exosomes and other small particles will elute together as they all will experience interactions with the pores in the column. The highest purity with SEC is achieved for sizes above 75 nm. SEC gives a great recovery of extracellular vesicles and a great removal of small contaminants [52]. For example, HDL would be isolated by SEC but not by density-gradient differential ultracentrifugation due to the density of HDL being similar to the density of EVs [45]. It is an inexpensive method that takes less than 20 min when compared to the up to 96 h required for ultracentrifugation. Furthermore, SEC, when paired with the density gradient technique (performed prior to SEC), leads to very high levels of purity for the cost and time to run the enrichment [62].

Asymmetric-flow field-flow fractionation (AF4), as shown in Figure 2H, separates nanoparticles based on their hydrodynamic sizes and density by two perpendicular flows, the laminar channel flow and the variable cross flow [63]. It is a powerful method that can separate nanoparticles with sizes ranging from nanometers to micrometers. AF4 starts with a sample that is injected into a stream with three flows: the laminar flow, the cross flow, and a flow opposing the laminar flow. The AF4 instrument is a rectangular box with an inlet, sample inlet, and channel outlet entering at the top. The smallest particles will elute first and upward due to their hydrodynamic size, causing diffusion up the column [64]. One study used this method and was able to separate exosome subpopulations by size as well as a nanoparticle in the cluster called exomeres, which differ in size and content [63] called AF4 “highly reproducible, fast, simple, label-free, and gentle”. The limitations of AF4 are size, sample amount, and prior isolation of EVs from the cell media. The AF4’s separability by hydrodynamic size is great, but it cannot select or differentiate different morphologies or surface molecules [63]. The AF4 can only hold small amounts of sample (~100 µg), and it requires isolation, usually ultracentrifugation, to concentrate the EVs [64]. Another method used is microfluidic devices, which are instruments that use very small amounts of fluid on a microchip to do certain laboratory tests (Figure 2I). They are classified into two different types: physical property-based microfluidic for exosome separation and immunoaffinity-based microfluidic for exosome separation [65].

Storage of EVs should be aliquoted because even one freeze-thaw cycle of EVs showed up to 10–15% losses in EV concentration [66]. Freezing is known to cause primarily vesicle rupture; the highest efficiency of preventing morphological differences in isolated EVs is a quick freeze at −80 °C. Notably, there is not enough research available on the depth of function loss of EVs due to freezing or prolonged storage [67]. Storage of extracellular vesicles in PBS at −80 °C was considered the status quo; however, new research claims that protein stabilizers like trehalose can reduce loss of EV concentration in freeze-thaw cycles as well as minimize increase in EV size distribution [68]. An in-depth buffer study suggests that the best combination out of 27 to store EVs used is PBS-HAT (PBS with HEPES, Albumin, and Trehalose) along with PBS-AT (Albumin and Trehalose), PBS-HATD (HEPES, Albumin, Trehalose and DMSO) in −80 °C [69].

## 4. Characterization

### 4.1. Morphometric Characterization

Microscopic evaluation and nanoparticle estimation techniques validate EVs and exosome morphology. Transmission Electron Microscopy (TEM) directs a high-energy electron beam through ultrathin samples stained with heavy metals like uranyl acetate to visualize EV ultrastructure. It resolves bilayer membranes and distinguishes exosomes (30–150 nm, cup-shaped under TEM), microvesicles (100–1000 nm), and apoptotic bodies (>1000 nm), as shown in studies of plasma EVs [70]. A practical example is [71], who used TEM to confirm exosome purity after ultracentrifugation, observing their characteristic morphology [72]. Figure 3A(1) depicts the use of TEM to visualize vesicles in the aqueous humor (AH) with the goal of comparing glaucomatous (AH) vesicles to healthy controls [73]. Atomic Force Microscopy (AFM), meanwhile, uses a nanoscale probe to map EV surface topography and mechanical properties, such as stiffness, which can indicate cargo packing density. For instance, Sharma et al. (2010) applied AFM to show softer EVs from cancer cells, suggesting altered lipid content affecting membrane fluidity in Figure 3A(2); a sample image of hydrated MCF-7 exosomes is shown, explaining more of the information provided by this method [74].

Cryo-electron microscopy (Cryo-EM) is another powerful technique used to visualize exosomes and other nanoparticles. In comparison to TEM and AFM, using cyro-EM allows for the sample to be preserved in its native state, maintaining biological structures at their natural state [80]. The imaging takes place at cryogenic temperatures (around −180 °C), which keeps the sample in place while minimizing damage to the sample. The high-resolution imaging provides a near-atomic resolution, allowing the detailed visualization of size, shape and surface characteristics. In Figure 3B(4), the image of an EV in platelet-rich plasma clearly shows the precision and quality of imaging that cryo-EM provides [79].

Nanoparticle Tracking Analysis (NTA) employs laser scattering to track EV Brownian motion in solution, calculating size via the Stokes-Einstein equation and concentration from particle counts (Figure 3B(1)) [53,75]. Its real-time analysis suits dynamic samples like a serum, where EV size shifts (e.g., increased microvesicles in inflammation) correlate with disease [62]. Ref. [81] used NTA to characterize Oxalobacter formigenes EVs, reporting a mean size of 122.9 ± 46.3 nm and a distribution (D10 = 80.4 nm, D50 = 111.5 nm, D90 = 182.6 nm), aligning with bacterial EV norms and validating their isolation. NTA’s quantitative precision complements MS by linking physical traits to molecular findings.

On the other hand, flow cytometry is used to characterize exosomes using fluorescent antibodies and other markers—not just analyze their size—but other properties, such as surface markers. After isolation and staining, the exosomes are injected into the flow chamber, and particles are passed through the laser chamber one at a time. During this time, the scattered light and fluorescence that is emitted will be detected by the sensors, which provide information used to analyze size distributions and properties. Figure 3B(2) provides an example plot from research performed with small extracellular vesicles (sEVS) [76]. Another technique is Dynamic Light Scattering (DLS), used to measure particle size and the release of exosomes from cells. Through the use of a light source, typically a laser, the relative sizes of the nanoparticles are obtained by analyzing the light intensity fluctuations created by the exosomes. As shown in Figure 3B(3), the diameter of a nanoparticle will be determined by the differential intensity fractionation as used in this research study referenced [78].

Raman Spectroscopy measures inelastic light scattering from molecular vibrations, producing spectra that fingerprint EV lipids (e.g., CH2 stretching at 2800–3000 cm^−1^), proteins (amide I at 1650 cm^−1^), and carbohydrates [54]. Ref. [82] used Raman to distinguish cancer EVs by elevated lipid signals, reflecting membrane changes. Fourier Transform Infrared Spectroscopy (FTIR), conversely, detects infrared absorption by functional groups, identifying protein secondary structures (amide I and II bands) and lipid acyl chains (CH2 at 2920 cm^−1^). [83] applied FTIR to bacterial EVs, noting shifts in protein bands linked to stress responses. Both methods, being label-free and non-destructive, offer rapid biochemical profiling, enhancing MS data interpretation. These techniques collectively bridge EV physical properties with their molecular cargo, ensuring a thorough analysis.

### 4.2. Compositional Characterization

The roles of exosomes in intercellular communication, disease progression, and therapeutic applications have spurred intense research, with mass spectrometry (MS) emerging as a critical tool for dissecting their molecular cargo [53,55]. Liquid Chromatography-Tandem Mass Spectrometry (LC-MS/MS) is a gold-standard technique for EV analysis due to its exceptional sensitivity and ability to characterize complex proteomic, metabolomics and lipidomics profiles within EV cargo (Figure 4) [84]. The process begins with liquid chromatography, where a sample is passed through a column (e.g., C18 reverse-phase) to separate peptides based on their hydrophobicity [85]. These separated molecules are then ionized via electrospray ionization (ESI-MS), producing charged species that are fragmented in a tandem mass spectrometer to generate detailed spectra [62]. For instance, ref. [84] used LC-MS/MS to identify over 1000 proteins in exosomes from dendritic cells, revealing markers like tetraspanins (CD9, CD63) linked to immune modulation.

This technique excels in uncovering functional pathways influenced by EVs. In neuroscience, LC-MS/MS has identified proteins such as synapsin and GAP43 in neural-derived EVs, implicating them in synaptic repair and axonal regeneration after traumatic brain injury [70]. The method’s versatility extends to quantitative analysis, using stable isotope labeling (e.g., SILAC) to compare protein abundance between healthy and diseased EV populations, making it a cornerstone for both discovery and translational research [6].

High-Resolution Mass Spectrometry (HRMS) enhances EV analysis by providing precise mass measurements, often to within 1–5 parts per million (ppm), using advanced instruments like Orbitrap and Time-of-Flight (TOF) analyzers. This precision allows researchers to resolve isobaric species—molecules with identical nominal mass but distinct exact masses—crucial for identifying subtle differences in EV cargo. HRMS employs two primary data acquisition strategies: Data-Dependent Acquisition (DDA) and Data-Independent Acquisition (DIA). DDA selects the most abundant precursor ions in each scan cycle for fragmentation, generating high-specificity tandem mass spectra. For example, Ref. [86] used DDA on an Orbitrap to profile EV proteins from mesenchymal stem cells, identifying growth factors like VEGF with high confidence. However, its focus on abundant ions can overlook low-level proteins, limiting its depth in complex samples. Data-independent acquisition fragments all ions within predefined mass windows, capturing a comprehensive snapshot of the proteome. Ref. [87] demonstrated DIA’s advantage by detecting low abundance signaling peptides in EVs missed by DDA, such as those involved in the Wnt pathway. This makes DIA particularly valuable for exhaustive EV profiling, especially in heterogeneous biological samples. In practice, combining DDA and DIA can maximize coverage [88]. A study of plasma-derived EVs used DDA to build a spectral library, followed by DIA to quantify over 2000 proteins, revealing biomarkers like apolipoproteins linked to cardiovascular disease [89]. HRMS thus offers a flexible, powerful framework for EV research.

Matrix-Assisted Laser Desorption/Ionization Time-of-Flight Mass Spectrometry (MALDI-TOF MS) provides a rapid, straightforward approach to EV analysis, ideal for high-throughput screening [90]. The technique involves mixing the sample with a matrix (e.g., sinapinic acid), which absorbs laser energy to ionize analytes, followed by mass measurement in a TOF analyzer. Its minimal sample preparation—often just spotting EVs onto a plate—enables quick profiling of proteins and peptides. For instance, MALDI-TOF MS distinguished EV protein signatures between Alzheimer’s patients and healthy controls, identifying amyloid-beta fragments as potential biomarkers [91].

In lipidomics, two approaches dominate EV studies: Shotgun Lipidomics and LC-MS-based Lipidomics [92]. Shotgun Lipidomics directly infuses lipid extracts into an MS system, bypassing chromatography, to detect species like phosphatidylcholines (PCs) and sphingomyelins (SMs) in a single run. Llorente et al. (2013) used this method to profile prostate cancer EVs, finding elevated SM levels associated with metastasis. LC-MS-based lipidomics, however, separates lipids first, offering a greater resolution for subclasses like phosphatidylethanolamines (PEs) and ceramides, which influence EV membrane stability and fusion [54]. For example, neural-derived EVs analyzed via LC-MS revealed cholesterol-rich profiles enhancing their uptake by astrocytes, a finding with implications for neuroregeneration [93]. These lipidomic insights complement proteomic data, linking EV structure to function.

### 4.3. Data Processing and Analysis in MS-Based Exosome Research

Mass spectrometry generates vast datasets, requiring robust bioinformatics for interpretation. In proteomics, raw data undergoes peak detection to identify mass-to-charge (*m*/*z*) signals, deconvolution to resolve overlapping peaks, noise filtration, and alignment to compare across samples. Tools like MaxQuant process these steps, matching spectra to databases like UniProt or SwissProt with algorithms that account for post-translational modifications (e.g., phosphorylation) and isoforms [94]. Quantification methods enhance this analysis: Label-Free Quantification (LFQ) uses peak intensities to compare protein levels, while Tandem Mass Tag (TMT) labeling enables multiplexed quantification, as seen in studies comparing EV cargo from tumor versus normal cells [95]. Table 2 references programs such as KEGG and ProteomeXchange, which are useful for pathway analysis and data repository of proteomics.

**Table 2 metabolites-15-00454-t002:** Data Analysis and Processing Resources.

Data Analysis and Processing Resources			
Omics Type	Data Analysis and Processing		Use(s)
Proteomics	Proteome Discoverer™	https://www.thermofisher.com/us/en/home/industrial/mass-spectrometry/liquid-chromatography-mass-spectrometry-lc-ms/lc-ms-software/multi-omics-data-analysis/proteome-discoverer-software.html?erpType=undefined (accessed on 9 April 2025)	Raw Data Processing, Data Normalization, Statistical Analysis, Identification of PTMs
	MaxQuant	https://www.maxquant.org/ (accessed on 9 April 2025)	Raw Data Processing
	Express Analyst	https://www.expressanalyst.ca/ (accessed on 9 April 2025)	Statistical Analysis, Data Visualization
	Uniprot	https://www.uniprot.org/ (accessed on 9 April 2025)	Protein and Peptide Description
	KEGG	https://www.genome.jp/kegg/ (accessed on 9 April 2025)	Pathway Analysis
	ProteomeXchange	https://www.proteomexchange.org/ (accessed on 9 April 2025)	Data Repository
Lipidomics	Lipid Search™	https://www.thermofisher.com/order/catalog/product/OPTON-30880 (accessed on 9 April 2025)	Raw Data Processing
	MS-DIAL	https://systemsomicslab.github.io/compms/msdial/main.html (accessed on 9 April 2025)	Raw Data Processing
	mzMine	https://mzio.io/mzmine-news/ (accessed on 9 April 2025)	Raw Data Processing
	Compound Discoverer™	https://www.thermofisher.com/us/en/home/industrial/mass-spectrometry/liquid-chromatography-mass-spectrometry-lc-ms/lc-ms-software/multi-omics-data-analysis/compound-discoverer-software.html (accessed on 9 April 2025)	Raw Data Processing, Spectral Identification
	MetaboAnalyst 6.0	https://www.metaboanalyst.ca/ (accessed on 9 April 2025)	Statistical Analysis, Data Visualization
	LipidOne	https://lipidone.eu/ (accessed on 9 April 2025)	Statistical Analysis, Data Visualization
	LIPEA	https://hyperlipea.org/home (accessed on 9 April 2025)	Pathway Analysis
	KEGG	https://www.genome.jp/kegg/ (accessed on 9 April 2025)	Pathway Analysis
	Metabolomics Workbench	https://www.metabolomicsworkbench.org/ (accessed on 9 April 2025)	Data Repository
Metabolomics	Compound Discoverer™	https://www.thermofisher.com/us/en/home/industrial/mass-spectrometry/liquid-chromatography-mass-spectrometry-lc-ms/lc-ms-software/multi-omics-data-analysis/compound-discoverer-software.html (accessed on 9 April 2025)	Raw Data Processing, Spectral Identification
	MS-DIAL	https://systemsomicslab.github.io/compms/msdial/main.html (accessed on 9 April 2025)	Raw Data Processing
	mzMine	https://mzio.io/mzmine-news/ (accessed on 9 April 2025)	Raw Data Processing
	MetaboAnalyst 6.0	https://www.metaboanalyst.ca/ (accessed on 9 April 2025)	Statistical Analysis, Data Visualization
	KEGG	https://www.genome.jp/kegg/ (accessed on 9 April 2025)	Pathway Analysis
	Metabolomics Workbench	https://www.metabolomicsworkbench.org/ (accessed on 9 April 2025)	Data Repository
Integromics	OmicsNet 2.0	https://www.omicsnet.ca/ (accessed on 9 April 2025)	Data Processing, Data Visualization
	OmicsAnalyst 2.0	https://www.omicsanalyst.ca/ (accessed on 9 April 2025)	Data Processing, Data Visualization

For lipidomics and metabolomics, Lipid Search, MS-DIAL, Compound Discoverer and others are commonly used for raw data processing. LipidONE, MetaboAnalyst 6.0, and metabolomics workbench provide tools for statistical analysis, data visualization and a widely used repository, respectively (more resources mentioned in Table 2). MetaboAnalyst integrates these data with pathway enrichment, linking EV metabolites like lactate to glycolysis in hypoxic tumors [96]. This multi-layered approach ensures comprehensive EV characterization. Additionally, integromics (the study that combines different omics) can be analyzed using the programs Omicsnet 2.0 and Omicsanalyst 2.0. This multi-layered approach ensures comprehensive EV characterization.

## 5. Discussion

Exosomes provide promise in the delivery of metabolites to promote axon regeneration when compared to the metabolite alone. Ref. [97] transported a neuropeptide, PACAP38, via exosomes into a rat with traumatic optic neuropathy. They used ultracentrifugation at 100,000× *g* to isolate the exosomes, transmission electron microscopy for morphology examination, and a western blot for biomarker analysis [97]. The rats underwent optic nerve crush as their form of traumatic optic neuropathy via a surgical procedure. They were assigned to treatment groups where they were treated with no injections or underwent injections with exosomes, exosomes with PACAP38, and PACAP38 alone. All models besides the untreated group experienced axon regeneration. Flash visual-evoked potential, or FVEPs, were used to capture the P2 wave to determine the improvement of optic nerve function [98]. FVEPs use a brief flash to measure the response of brain electrical activity, which indicates early cortical processing [99]. The P2 is the second positive wave of the FVEP [100]. Seven days post-injury, the latency of the P2-wave was preserved in the exosome PACAP38 treatment, where the other treatments matched the control [97]. Fourteen days after injury showed significant preservation of the P2-wave latency preservation, which confirmed the potency of exosome delivery in improving optic nerve function. Wang and colleagues claim that there were no drug-related adverse effects of exosome delivery [97].

Exosomes from other cells have notably influenced axon regeneration. Mesenchymal stem cells are stromal cells that can self-renew and exhibit multilineage differentiation [101]. Specifically, the umbilical cord mesenchymal stem cells (UMSCs) have had their exosomes isolated by differential centrifugation and then delivered surgically to mice [102]. Mice were compared using an optic nerve crush on one nerve, and exosomes were delivered intravitreally. Ref. [102] noted that UMSCs-exosomes promoted the survival of RGCs but did not promote optic nerve axon regeneration. They delineated this by counting the proportion of RGCs and Brn3a, a transcription factor crucial for neuronal development and function, in the retinas of optic nerve crush retinas against controls. Brn3a is a reliable, efficient marker to quantify RGCs in retinas [103]. Furthermore, UMSCs-exosomes enhanced glial activity. The scientists proposed two ideas to the results obtained: enhanced glial activity could contribute to the survival of RGCs by secreting neurotrophins, and/or the specific miRNAs in UMSCs-exosomes promoted survival of RGCs. Specifically, miR-21 was the most abundant miRNA in UMSCs-exosomes, and the function of that specific miRNA could be to over-activate the astrocytes. The exosomes were delivered to the mice intravitreally on days 0, 7, and 14, with sacrifice on day 21 [102]. The delivery was performed with a 33 g Hamilton syringe where a 5 µL volume of sPBS loaded with 1 × 10^9^ exosomes was injected slowly with needle retraction after 2 min for backflow reduction [104]. Both groups studying MSC exosomes noted their integration into both neurons and astrocytes, providing unclear direction as to whether the glial cells or RGCs caused the therapeutic effect. A similar method was used on bone marrow mesenchymal stem cells (BMSCs) and fibroblasts, where their exosomes were isolated and delivered intravitreally to mice with ocular nerve crush [104]. Ref. [104] found that BMSC exosomes provided significant neuroprotection to RGC axons by OCT and ERG data. Furthermore, over 50% of RGC function was maintained, so the exosomes not only kept the RGCs from dying but also fully functional [104]. The regeneration of RGCs with BMSC-exosomes was only significant at shorter distances from the lesion site, limiting its potential to aid in long-distance axon regeneration [104]. The reasoning behind the isolation of mesenchymal stem cell exosomes is due to their previously known axonal regenerative effects when delivered intravitreally in cellular form without exosome purification and isolation [105,106,107]. Another study demonstrated that hypoxia-stimulated astrocytes secreted exosomes containing miR-329-5p, which protected RGCs from apoptosis through inhibition in the JNK pathway responsible for RGC death after axonal injury. While miRNA identification has been successful, there remains much to be researched as far as metabolites and lipids are concerned with exosome mediation therapies in axon regeneration. Spatial lipidomics is an emerging field, although sparse so far in CNS research [108]. However, it has been applied to investigate the optic nerve lipids as well as demyelination/remyelination of the spinal cord [108,109,110,111]. Subsequent extractive lipidomics for optic nerve [110] have confirmed findings of lipid species using spatial omics [109]. It must be borne in mind that it is quite challenging to obtain pure exosome preparations using the methods currently employed (even with the application of a combination of methods), and for this reason, several scientists working in this area now prefer the term “microvesicles” for these preparations.

Metabolites are key contributors to integrating pathways that allow for axon regeneration [112]. In *Drosophila*, reprogramming glial cells has been proven to promote axon regeneration [113]. Ref. [113] activated the PI3K and EGFR pathways in glial cells and found extensive regrowth in the neuropil region with a regeneration percentage greater than 90% in 24 h. Indole-3, a gut metabolite, promotes axonal regeneration after sciatic nerve crush in mice [114]. While the sciatic nerve is a peripheral nerve, it is important to note that metabolic processes can change and enhance axon regeneration through metabolic pathway mechanisms. Peptides like the CH02 peptide serve as metabolites that have been proven to promote neurite growth in rats by inducing axon regeneration through FGFR and AKT/mTOR signaling [115]. In Table 3, metabolites that have been found to play a role in CNS and PNS regeneration have been highlighted [38,92,116,117,118,119,120,121,122,123,124,125,126,127,128,129,130,131,132,133,134,135]. These metabolites are also found in microvesicles and exosomes, making their role as key contributors in the interaction of mechanistic pathways in axon regeneration clear. Some of the key metabolites are Allantoin, Taurine, and lipids such as Gangliosides, Lysophosphatidylcholine (LPC), Lysophosphatidylethanolamine (LPE) and Lysophosphatidic acid (LPA).

**Table 3 metabolites-15-00454-t003:** Metabolites in Microvesicles and Exosomes.

Microvesicles		
Metabolites	Axon Regeneration	Reference (PMIDs)
*N*-acetylglycosamine	CNS, PNS	[116,117]
Allantoin	PNS	[118,136]
Histidine	CNS, PNS	[119]
Lysophosphatidylcholine (LPC)	CNS	[120,121]
Lysophosphatidylethanolamine (LPE)	CNS	[122,123]
Taurine	CNS	[124,125,126,127]
Gangliosides	CNS, PNS	[38]
**Exosomes**		
**Metabolites**	**Axon regeneration**	**Reference**
Acetyl-L-Carnitine (ALCAR)	CNS, PNS	[128,129]
Carnosine	CNS	[130,131]
Dimethylglycine	CNS	[132]
Histidine	CNS, PNS	[119]
Lysophosphatidylcholine (LPC)	CNS	[120,133]
Lysophosphatidylethanolamine (LPE)	CNS	[122,123]
Taurine	CNS	[124,125,126,127]
Gangliosides	CNS, PNS	[38]
Lysophosphatidic acid (LPA)	CNS, PNS	[134]
**Modified vesicles/exosomes**		
**Metabolites**	**Axon regeneration**	**Reference**
Sphingolipids	CNS	[92,135]

Given both the knowledge of axon regeneration being promoted by metabolites due to their influence on promoting metabolic pathways that enhance axon regeneration and the knowledge that exosomes are highly biocompatible and small to deliver drugs, we propose a combination of the two could significantly enhance the efficacy of treatments and potency of the axon regeneration.

## 6. Conclusions

Extracellular vesicles’ therapeutic properties for axonal regeneration are promising. With further research, having a comprehensive knowledge of morphometric and compositional characterization will assist with learning how EVs can be used in future therapeutics. While there is extensive research still to be performed, by standardizing techniques such as isolation, characterization, and packaging, practices can be established in hopes of possible therapies. With these advancements and learning more about the metabolites that are crucial for axonal regeneration, extracellular vesicles are promising to advance the field.

## Figures and Tables

**Figure 1 metabolites-15-00454-f001:**
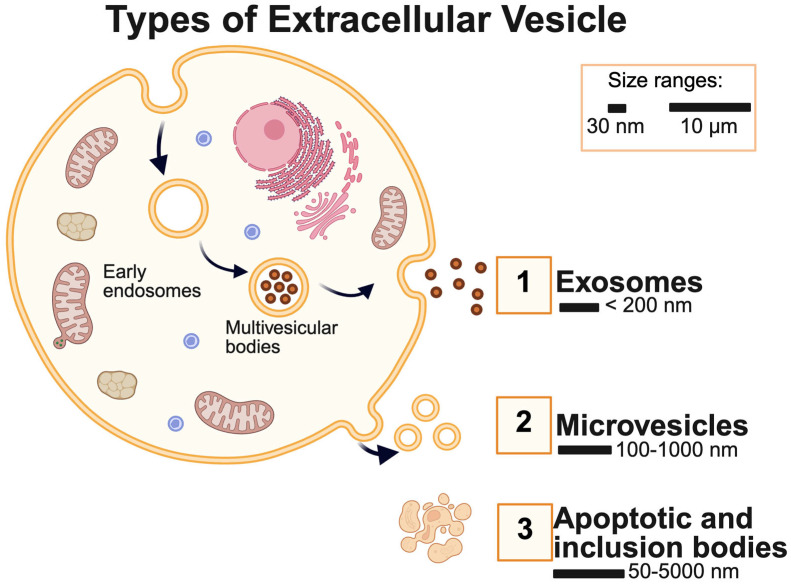
**Types of Extracellular Vesicles (EVs).** The sizes of EVs range from 30 nm to 10 µm. 1. Exosomes are usually smaller than 200 nm, while 2. Microvesicle sizes range from 100 to 1000 nm. 3. Apoptotic and inclusion bodies are also components of extracellular vesicles. These are the most common and historically well-characterized subtypes. The field is rapidly evolving, and new vesicle types continue to be identified.

**Figure 2 metabolites-15-00454-f002:**
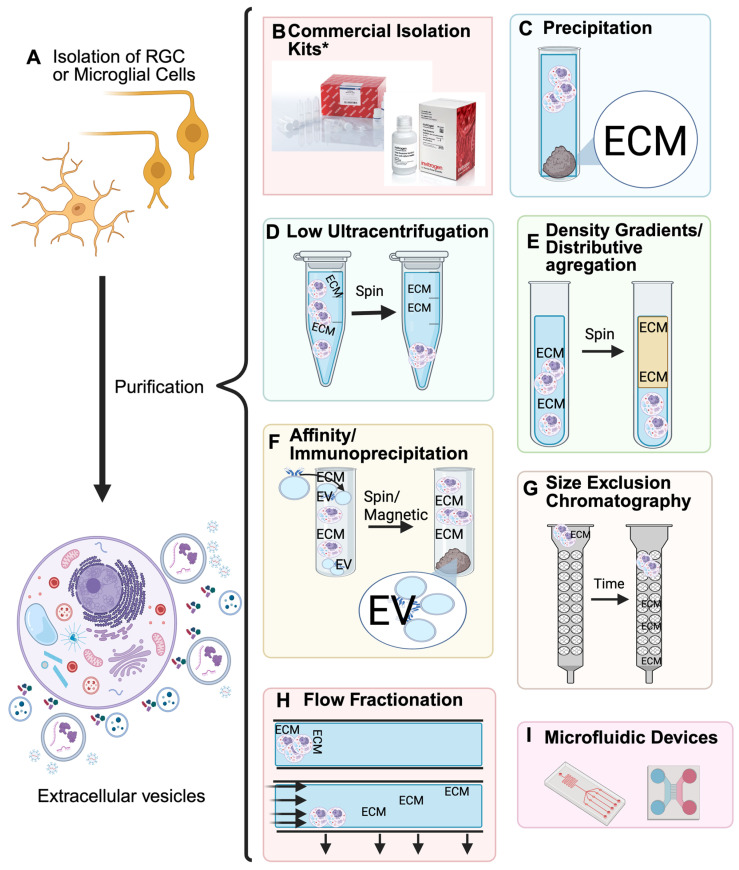
**The isolation of Extracellular Vesicles (EVs) from RGCs and MGCs**. (**A**) With **commercial isolation kits,** EVs will be isolated from the RGCs and MGCs; some examples are the “Total isolation kit” from ThermoFisher and “ExoQuick Exosome and RNA Purification Kit by System Bioscience (more information on isolation kits is given in Table 1). The ECM consists of extracellular proteins, extracellular vesicles, exosomes, and other extracellular particles. (**B**) Several commercially available kits enable isolation of exosomes or microvesicle. They use propreiatary materials to isolate exosomes/microvesicles. * Table 1 gives reference to some of the commercially available isolation kits. (**C**) **Precipitation** uses substances that wrap and aggregate the ECM from the cells Subsequent to usage of materials centrifugation is used to generate a pellet. While it is simple to use and has a high yield of particles, the purity of samples is low, and this can interfere with downstream analyses. (**D**) **Ultracentrifugation** is used at different rates to isolate the largest structure (cells and apoptotic bodies) and to isolate the ECM. While this is the gold standard to separate extracellular vesicles from diverse cell media, it is challenging to separate by size, which is why a combination of methods is recommended (**E**) **Density Gradients** use density and solubility properties to aggregate EVs from cells and other impure substances. This is best used when particles have distinct density values, but if similar in composition, this method will have low purity. (**F**) **Affinity/Immunoprecipitation** uses magnetic or antibody properties to isolate desired EVs by centrifuge or magnet. While this method has a high specificity and purity, some EV markers might not be fully individualized and not specifically bind the desired target. (**G**) **Size Exclusion Chromatography** is a column that elutes smaller particles (ECM) faster than larger particles, isolating EVs among other extracellular particles from RGCs and MGCs. This technique is highly inexpensive, less time-consuming than others, and obtains high purity of samples. However, if column parameters are not optimized, there might be a lower yield of samples. (**H**) **Flow Fractionation**: using particle size to separate cell culture with the largest particles moving the slowest along a fraction. This method is effective in distinguishing between EVs subpopulations accurately; however, it often needs validation from other techniques to verify the identity of EVs. (**I**) **Microfluidic devices** can move the cell culture along a path, moving smaller particles faster than larger ones isolating the ECM. While this technique offers high specificity and sensitivity with a low sample volume required, there is limited standardization, and the device’s channels can become clogged due to debris, which affects the sample’s accuracy.

**Figure 3 metabolites-15-00454-f003:**
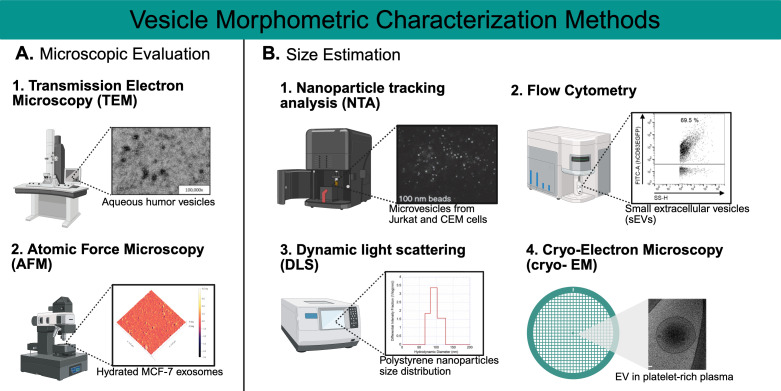
**The vesicle morphometric characterization methods.** (**A**) Microscopic Evaluation Methods. (1) Transmission Electron Microscopy (TEM) offers an image of small particles such as aqueous humor vesicles [72]. TEM allows for precise size measurements, especially with small EVs, which might not be identified with other techniques. (2) Atomic Force Microscopy (AFM) provides the three-dimensional geometry, size and other physical properties of exosomes; in this example, we see hydrated MCF-7 exosomes [74]. (**B**) Size estimation methods (1) Nanoparticle tracking analysis (NTA) allows the individual visualization and counting of exosomes and microvesicles according to their sizes [75,76]. (2) Flow cytometry, through the analysis of a single vesicle, we can study the heterogeneity of extracellular vesicles [77]. (3) Dynamic light scattering (DLS) is a technique used to measure nanoparticle sizes; as per the example shown, this method will allow the size distribution of nanoparticles [78]. (4) Cryo-electron microscopy allows the visualization of vesicles in their native state through the process of rapidly freezing them; in the example shown, we can see an illustration of EV in plasma [79].

**Figure 4 metabolites-15-00454-f004:**
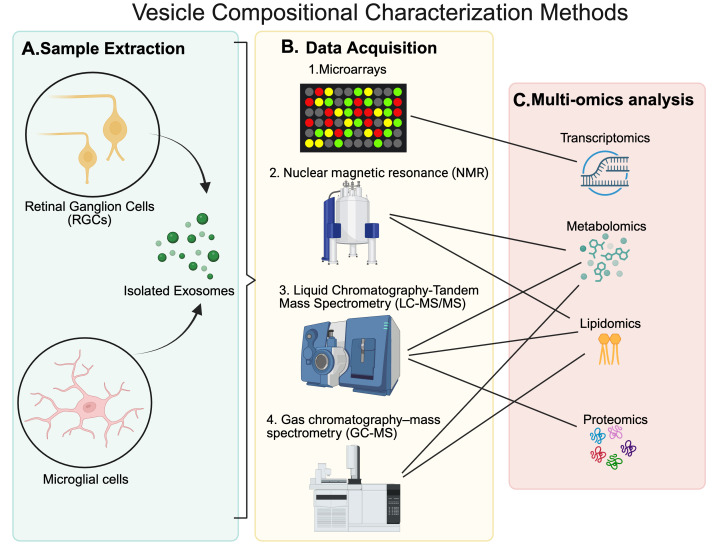
**The vesicle compositional characterization methods.** (**A**) Sample extraction from retina ganglion cells (RGCs) and microglial cells is performed through the methods shown in the previous figures, and kits allow exosomes to be isolated for data collection. (**B**) Data acquisition will be made through a variety of methods such as microarrays, RNA sequencing (RNA-seq), nuclear magnetic resonance (NMR), gas chromatography-mass spectrometry (GC-MS) and liquid chromatography-tandem mass spectrometry (LC-MS/MS). These different methods often obtain complementary information; for example, microarray and RNA-seq provide transcriptomics data complementing proteomics obtained through LC-MS/MS. Many other methods, such as protein array, Western blot etc., enable obtaining complementary as well as confirmatory information. For example, proteomics data obtained using LC-MS/MS may be confirmed using Western blot and/or protein array analysis. (**C**) Multi-omics analysis is possible based on the data acquired in transcriptomics, metabolomics, lipidomics, and proteomics analyses. Comprehensive application of analytical methods is needed for full characterization of exosome composition.

## Data Availability

No new data were created or analyzed in this study.

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
