# Peer review of "Extracellular Vesicle Metabolomics Holds Promise for Adult Axon Regeneration"

_metabolites, 2025, doi:10.3390/metabo15070454_

Round 1
Reviewer 1 Report
Comments and Suggestions for Authors
The current review is within the topic of EVs use for adult axon regeneration. While the field is of interest to the research community, the review is not focused on the topic referred in the title and has an odd structure, which limits the impact of the paper. Specific comments are as follows:
- The main issue is how the paper is structured, which poorly relates to the message that the abstract and title want to send. While section 1 and 2 introduce and summarize axon regeneration and EVs, sections 3 and 4 are limited to describe a series of techniques to isolate and characterize EVs in a very superficial manner to EVs in CNS and PNS. A deeper dive into their use and studies on EVs from CNS and PNS should be included, rather than putting too much effort into describing the techniques themselves.
- The title is very confusing. Metabolomics is the analysis of metabolites within EVs (in this case), how does it relate to adult axon regeneration? As a diagnostic? As characterization of EV before use as therapeutic? Also, replacement of promise by potential is recommended.
- Why section 3 is focused on retinal ganglion cells? Authors introduce first the potential of MSCs and Neural progenitor cells as source of cells for EV production. This seems like cherry picking in an area where authors feel comfortable with, but other sources of cells should be discussed. Also, the isolation is way too detailed for the aim of the review
- Figure 2, 3, and 4 are very generalist. Are these just separate / discrete techniques? Are they sequential? Also pros and cons of each for the specific use in EVs (and, if possible in axon regeneration) should be enumerated, helping the reader to understand how these techniques may be used in their research.
- Similarly for table 1, if no further information on pros and cons is included, this feels like advertisement for the products. Yield? Most used for? References?
- Discussion section is right on point for what the paper is trying to transmit. This section should be developed and a more studies in CNS and PNS using EVs should be included, by relating those to metabolomics research used in the field
Minor comments:
- Line 41 reads exomes
- RGCs no introduced before
- In the introduction of IA: markers are not definitive nor consistent within EV populations (i.e., not all EVs present the same markers pool), and this limitation should be noted
- AF4 is overexplained for the goal of the paper, and with very low clarity. Also, text reads [Depicted at FFF theory] which is not very clear what it means
Author Response
We thank the reviewer for his/her encouraging comments. Our responses are provided in blue fonts below every comment.
Briefly, we have addressed the comments to the best of our abilities by making appropriate and commensurate changes to the manuscript text. We have revised and expanded legend descriptions for figures 2, 3 and 4, and revised table 1 as per reviewers’ recommendations. We have revised the title, various sections and have checked the grammar throughout the manuscript text.
Reviewer #1
The current review is within the topic of EVs use for adult axon regeneration. While the field is of interest to the research community, the review is not focused on the topic referred in the title and has an odd structure, which limits the impact of the paper. Specific comments are as follows:
- The main issue is how the paper is structured, which poorly relates to the message that the abstract and title want to send. While section 1 and 2 introduce and summarize axon regeneration and EVs, sections 3 and 4 are limited to describe a series of techniques to isolate and characterize EVs in a very superficial manner to EVs in CNS and PNS. A deeper dive into their use and studies on EVs from CNS and PNS should be included, rather than putting too much effort into describing the techniques themselves.
We have revised our isolation and characterization methods as recommended.
- The title is very confusing. Metabolomics is the analysis of metabolites within EVs (in this case), how does it relate to adult axon regeneration? As a diagnostic? As characterization of EV before use as therapeutic? Also, replacement of promise by potential is recommended.
We have revised the title as recommended.
- Why section 3 is focused on retinal ganglion cells? Authors introduce first the potential of MSCs and Neural progenitor cells as source of cells for EV production. This seems like cherry picking in an area where authors feel comfortable with, but other sources of cells should be discussed. Also, the isolation is way too detailed for the aim of the review
We thank the reviewer for his/her comments. We have revised as recommended.
- Figure 2, 3, and 4 are very generalist. Are these just separate / discrete techniques? Are they sequential? Also pros and cons of each for the specific use in EVs (and, if possible in axon regeneration) should be enumerated, helping the reader to understand how these techniques may be used in their research.
Briefly, they are separate techniques. They have been written in a generalized form for the usage by a wide range of researchers engaged in axon regeneration (optic nerve, spinal cord) as well as for the researchers in neurodegenerative diseases. We have expanded the figure legends to include pros and cons as suggested by the reviewer.
- Similarly for table 1, if no further information on pros and cons is included, this feels like advertisement for the products. Yield? Most used for? References?
We have made every effort to substantially improve Table 1 during this revision in light of the reviewer’s comments.
- Discussion section is right on point for what the paper is trying to transmit. This section should be developed and a more studies in CNS and PNS using EVs should be included, by relating those to metabolomics research used in the field.
We thank the reviewers for his/her comments, our extensive search yielded limited published studies in this emerging area which has been cited.
Minor comments:
- Line 41 reads exomes
- RGCs no introduced before
- In the introduction of IA: markers are not definitive nor consistent within EV populations (i.e., not all EVs present the same markers pool), and this limitation should be noted
- AF4 is overexplained for the goal of the paper, and with very low clarity. Also, text reads [Depicted at FFF theory] which is not very clear what it means.
We have revised these texts as recommended. In light of the reviewer’s criticism some sentences have been removed in the revised text including AF4 over explanation and FFF theory.

Reviewer 2 Report
Comments and Suggestions for Authors
Overall, this review paper on extracellular vesicle metabolomics for axon regeneration is a scholarly and valuable contribution, and pending some minor corrections, should be published.
Only one major point for the authors to consider, and one that ties MALDI to lipidomics in general, is the field of spatial lipidomics. Although it is a review paper itself, the following review on spatial lipidomics is a valuable one that speaks to the enormous potential of this technique to axon regeneration:
Jha D, Blennow K, Zetterberg H, Savas JN, Hanrieder J. Spatial neurolipidomics—MALDI mass spectrometry imaging of lipids in brain pathologies. J Mass Spectrom. 2024; 59(3):e5008. doi:10.1002/jms.5008
There is an opportunity in this review, perhaps in the Discussion section, to note that spatial lipidomics is an emerging field, although sparse so far in CNS research. It has been applied at least once to investigate demyelination/remyelination of spinal cord:
Sekera ER, Saraswat D, Zemaitis KJ, Sim FJ, Wood TD. MALDI Mass Spectrometry Imaging in a Primary Demyelination Model of Murine Spinal Cord. J Am Soc Mass Spectrom. 2020 Dec 2;31(12):2462-2468. doi: 10.1021/jasms.0c00187.
The remainder of my comments are minor of an editorial nature and listed below.
- line 49. The sentence starting with "Providing..." is actually a sentence fragment. It can be tied to the previous sentence though to make it grammatically correct.
- line 234. A space is needed between "the" and "small"
- line 236. Instead of stating recovery is great, I would supply a value indicated in ref. 44.
- line 238. A space is needed between "to" and "96"
- line 249. Many years ago, a prominent analytical chemist instructed my classmates and myself to never use the term "machine" when referring to an analytical "instrument" as it is too imprecise.
- Figure 3 legend. The citations in the legend do not correspond in most of the cases to the citatation number in the Figure itself; I would recommend deleting all numeric refences in the figure graphic and leaving them in the legend.
- line 319. Instead of "en" the intention seems to be "an"
- line 320. The letter -o is missing from cryo-
Author Response
We thank the reviewer for his/her encouraging comments. Our responses are provided in bold blue fonts below every comment.
Briefly, we have addressed the comments to the best of our abilities by making appropriate and commensurate changes to the manuscript text. We have revised and expanded legend descriptions for figures 2, 3 and 4, and revised table 1 as per reviewers’ recommendations. We have revised the title, various sections and have checked the grammar throughout the manuscript text.
Reviewer #2:
Overall, this review paper on extracellular vesicle metabolomics for axon regeneration is a scholarly and valuable contribution, and pending some minor corrections, should be published.
Only one major point for the authors to consider, and one that ties MALDI to lipidomics in general, is the field of spatial lipidomics. Although it is a review paper itself, the following review on spatial lipidomics is a valuable one that speaks to the enormous potential of this technique to axon regeneration:
Jha D, Blennow K, Zetterberg H, Savas JN, Hanrieder J. Spatial neurolipidomics—MALDI mass spectrometry imaging of lipids in brain pathologies. J Mass Spectrom. 2024; 59(3):e5008. doi:10.1002/jms.5008
There is an opportunity in this review, perhaps in the Discussion section, to note that spatial lipidomics is an emerging field, although sparse so far in CNS research. It has been applied at least once to investigate demyelination/remyelination of spinal cord:
Sekera ER, Saraswat D, Zemaitis KJ, Sim FJ, Wood TD. MALDI Mass Spectrometry Imaging in a Primary Demyelination Model of Murine Spinal Cord. J Am Soc Mass Spectrom. 2020 Dec 2;31(12):2462-2468. doi: 10.1021/jasms.0c00187.
We thank the reviewer for his/her valuable advice and suggestions. We have revised these texts as recommended and cited these references in the discussion.
The remainder of my comments are minor of an editorial nature and listed below.
- line 49. The sentence starting with "Providing..." is actually a sentence fragment. It can be tied to the previous sentence though to make it grammatically correct.
- line 234. A space is needed between "the" and "small"
- line 236. Instead of stating recovery is great, I would supply a value indicated in ref. 44.
- line 238. A space is needed between "to" and "96"
- line 249. Many years ago, a prominent analytical chemist instructed my classmates and myself to never use the term "machine" when referring to an analytical "instrument" as it is too imprecise.
- Figure 3 legend. The citations in the legend do not correspond in most of the cases to the citatation number in the Figure itself; I would recommend deleting all numeric refences in the figure graphic and leaving them in the legend.
- line 319. Instead of "en" the intention seems to be "an"
- line 320. The letter -o is missing from cryo-
We thank the reviewer for his/her comments. We have revised as recommended incorporating all these changes.

Round 2
Reviewer 1 Report
Comments and Suggestions for Authors
To the view of this reviewer, the efforts to improve the manuscript are still insufficient and further edits are required.
1) Comment 1 has not been addressed. How do sections 3 and 4 relate to first sections? The use of different cells is not well linked nor rationalized. In each case, how do previous studies perform? What are the disadvantages and advantages of each? If the reader had to make a choice for their research, what should they consider?
2) Title: Authors just changed "promise" to "potential". What is the final goal of using metabolomics? Diagnostic tool? Characterization prior use as therapeutic? Quality assurance? Other?
3) Same as comment 1. Poor linking and rationale behind the use of different cells
4) Could the table 1 be further improved by including other interesting outputs? Yield? Limitations?
5) Discussion comment: If the literature has a gap in this field, this should be noted in the discussion in a new paragraph while pointing other potential limitations in the field
6) In the response letter to reviewer, please indicate exactly where in the text each comments is addressed
Author Response
To the view of this reviewer, the efforts to improve the manuscript are still insufficient and further edits are required.
Response: We profusely thank the reviewer for his/her comments. We apologize for unsatisfactory changes in the last round. We now have made an earnest effort to address these comments. We are immensely grateful to this reviewer for his/her scrutiny and helpful comments to improve our manuscript.
- Comment 1 has not been addressed. How do sections 3 and 4 relate to first sections? The use of different cells is not well linked nor rationalized. In each case, how do previous studies perform? What are the disadvantages and advantages of each? If the reader had to make a choice for their research, what should they consider?
Response: We thank the reviewer for his/her comments. We have revised page 4, line 129-133, page 6, line 186-188, line 198-200, page 7 table 1 footnote, page 9, line 291-293, page 11, line 366-369, page 12, line 433-440 has been added/modified in response to reviewer’s comments.
- Title: Authors just changed "promise" to "potential". What is the final goal of using metabolomics? Diagnostic tool? Characterization prior use as therapeutic? Quality assurance? Other?
Response: We thank the reviewer for his/her comments. Page 14, line 464-469, page 15, line 520-523, Page 17, line 548-557 have been added/revised in response to the reviewer’s comments.
3) Same as comment 1. Poor linking and rationale behind the use of different cells
Response: We thank the reviewer for his/her comments. The revisions changes made in response to this comment is also capture in response to comment 1 above. page 12, line 433-440 has been added/modified in response to reviewer’s comments.
4) Could the table 1 be further improved by including other interesting outputs? Yield? Limitations?
Response: Page 7, table 1 footnote has been revised in response to these comments.
5) Discussion comment: If the literature has a gap in this field, this should be noted in the discussion in a new paragraph while pointing other potential limitations in the field
Response: We thank the reviewer for his/her comments. Page 14, line 464-469, page 15, line 520-523, Page 17, line 548-557 have been added/revised in response to the reviewer’s comments.
6) In the response letter to reviewer, please indicate exactly where in the text each comments is addressed
Response: We have noted page numbers in all our responses above. We also have highlighted all changes in the manuscript document.